# Disabled-2 (*DAB2*): A Key Regulator of Anti- and Pro-Tumorigenic Pathways

**DOI:** 10.3390/ijms24010696

**Published:** 2022-12-31

**Authors:** Zoe K. Price, Noor A. Lokman, Masato Yoshihara, Hiroaki Kajiyama, Martin K. Oehler, Carmela Ricciardelli

**Affiliations:** 1Discipline of Obstetrics and Gynaecology, Robinson Research Institute, Adelaide Medical School, University of Adelaide, Adelaide, SA 5005, Australia; 2Department of Obstetrics and Gynecology, Nagoya University Graduate School of Medicine, Nagoya 464-0813, Japan; 3Department of Gynaecological Oncology, Royal Adelaide Hospital, Adelaide, SA 5000, Australia

**Keywords:** *DAB2*, cancer, metastasis, TGFβ, Wnt, MAPK

## Abstract

Disabled-2 (*DAB2*), a key adaptor protein in clathrin mediated endocytosis, is implicated in the regulation of key signalling pathways involved in homeostasis, cell positioning and epithelial to mesenchymal transition (EMT). It was initially identified as a tumour suppressor implicated in the initiation of ovarian cancer, but was subsequently linked to many other cancer types. *DAB2* contains key functional domains which allow it to negatively regulate key signalling pathways including the mitogen activated protein kinase (MAPK), wingless/integrated (Wnt) and transforming growth factor beta (TGFβ) pathways. Loss of *DAB2* is primarily associated with activation of these pathways and tumour progression, however this review also explores studies which demonstrate the complex nature of *DAB2* function with pro-tumorigenic effects. A recent strong interest in microRNAs (miRNA) in cancer has identified *DAB2* as a common target. This has reignited an interest in *DAB2* research in cancer. Transcriptomics of tumour associated macrophages (TAMs) has also identified a pro-metastatic role of *DAB2* in the tumour microenvironment. This review will cover the broad depth literature on the tumour suppressor role of *DAB2*, highlighting its complex relationships with different pathways. Furthermore, it will explore recent findings which suggest *DAB2* has a more complex role in cancer than initially thought.

## 1. Introduction

Disabled-2 (*DAB2*) is a widely recognised tumour suppressor. It was initially discovered in 1994 when Mok et al. identified an 800bp cDNA fragment which was expressed in normal ovarian surface epithelial cell lines but not in ovarian cancer cell lines. They referred to it as differentially expressed in ovarian carcinoma 2 (DOC-2) [1]. The following year, Xu et al. identified a 96 kDa phosphoprotein in mouse macrophage cell line, BAC1.2F5 with an amino terminal end which shared homology to the Drosophila disabled gene [2,3]. The Drosophila disabled protein is important in embryogenesis and neural positioning [4]. *DAB2* is one of two human orthologs of the Drosophila disabled gene, Disabled-1 is expressed almost exclusively in neural cells whereas *DAB2* is expressed in a wide range of epithelial cells including those of the ovary, lung and breast [5,6,7]. Loss of *DAB2* expression has been reported in a range of malignancies including ovarian, lung and breast cancer [5,6,7] Table 1. The loss of *DAB2* is associated with activation of key signalling pathways including Wnt, MAPK and TGFβ which is associated with enhanced cell proliferation, chemotherapy resistance and tumour progression, supporting its role as a tumour suppressor. This review will discuss the role of *DAB2* in regulating these key pathways and the resulting effects on cancer progression.

## 2. *DAB2* Structure and Function

The human *DAB2* gene, located on chromosome 5p13 consists of 15 exons, encoding a 770 amino acid protein [8]. The mouse *DAB2* gene has 83% homology with the human gene, it also consists of 15 exons and encodes a 766 amino acid protein [9]. There are two isoforms of *DAB2*, including full length p96 (also known as p82) and spliced p67 (also known as p59) that is missing the central exon. *DAB2* contains binding domains and motifs which allow it to recognise and recruit proteins to clathrin coated pits for endocytosis (Figure 1). Two key binding domains of *DAB2* are a phosphotyrosine binding (PTB) domain at the N-terminus and a proline rich domain (PRD) at the carboxy terminal end of the protein which contains a myosin interacting region (MIR). The main function of *DAB2* is as a clathrin associated sorting protein (CLASP) in clathrin mediated endocytosis. *DAB2* interacts with clathrin via multiple binding sites including a type I LVDLN and type II PWPYP sequence [10]. *DAB2* can interact with both pre-assembled clathrin cages and also soluble clathrin trimers, indicating a possible role in clathrin cage assembly [10]. The *DAB2* PTB binds to phosphoinositide(4,5)P2 (PtdIns(4,5)P2) containing liposomes, further suggesting it is involved in clathrin cage assembly and vesicle budding [10].

The principle and first adaptor protein identified for clathrin mediated endocytosis (CME) was AP-2 tetramer, which recognises the YXXØ motif of target receptors. Two receptors which undergo CME, low density lipoprotein receptor (LDLR) and epidermal growth factor receptor (EGFR) cannot interact with AP-2 as they lack the YXXØ motif, however, both receptors contain FxNPxY motifs [11]. *DAB2* is involved in the internalisation of both LDLR and EGFR through interactions between its PTB domain and their FxNPxY sequence [11,12]. The recruitment of LDLR via *DAB2* occurs independent of AP-2 and ARH (LDLR adaptor protein) but via its interaction with clathrin and PtdIns(4,5)P2 [12]. Full length p96 *DAB2* also contains DPF motifs within the central exon which interact with the ⍺-adaptin subunit of AP-2 [11]. *DAB2* co-localises with AP-2 and LDLR in clathrin coated pits and early endosomes. In the cell *DAB2* dissociates from LDLR before it reaches late endosomes or lysosomes [11].

The MIR domain of *DAB2*, spanning amino acids 675-713, contains two functional motifs ^682^SYF^684^ and ^699^DFD^701^ [13]. The SYF motif is required for binding to the myosin VI cargo binding domain (CBD) [13]. The DFD motif also interacts with the myosin VI CBD, however this interaction induces chemical changes within the myosin VI structure [13]. These interactions promote the homodimerisation of myosin VI which then transports clathrin coated vesicles throughout the cell along actin networks [14,15]. This interaction between *DAB2* and myosin VI is dynamic allowing the transport of clathrin coated vesicles throughout the dense actin networks with minimal disruption to the actin fibres [14]. *DAB2* has also been implicated as a negative regulator of myosin VI nuclear activity, such as the transcription of oestrogen receptor (ER) target genes in MCF-7 breast cancer cells [16].

*DAB2* has also been shown to be involved in immune regulation. A review by Figliuolo da Paz et al. extensively explores the roles of *DAB2* in immune regulation in both innate and adaptive immune responses [17].

*DAB2* expression in antigen presenting cells (APCs) is downregulated during inflammation [17]. Under normal homeostasis, *DAB2* expression is activated by binding of Ets- transcription factor and PU.1 to the *DAB2* promoter [18]. During inflammation, interferon gamma (INF-γ), activates downstream transcription factor, interferon consensus sequence binding protein (ICSBP), which competes for binding to the promoter, inhibiting *DAB2* expression [18]. *DAB2* promotes cell spreading of RAW264.1 macrophage cell lines and enhances adhesion to ECM components collagen IV and laminin [18]. *DAB2* regulates switching from a pro-inflammatory M1 macrophage phenotype to a M2 phenotype which promotes tissue repair and reduces inflammation [19]. *DAB2* interacts with tumour necrosis factor receptor associated factor 6 (TRAF6) via 2 domains at aa226 and aa689, preventing activation of Nuclear factor kappa B (NF-κB) and subsequent expression of pro-inflammatory genes in M1 macrophages [19]. Loss of *DAB2* was required for pro-inflammatory responses to Toll-like receptor (TLR) ligands lipoteichoic acid (LTA) and lipopolysaccharide (LPS) [19,20]. *DAB2* is highly expressed on CD11b^+^CD103^−^ dendritic cells (DC) which are involved in Th17 and Th1 responses in the gut [21]. Loss of *DAB2* enhanced colitis in mouse models suggesting a role of *DAB2* in immune tolerance [21]. TLR4 ligand LPS downregulates *DAB2* expression and shifts the DCs to a mature, activated DC [21]. Furthermore, *DAB2* silencing in bone marrow derived DC activates PI3K and NF-κB, enhancing the expression of pro-inflammatory cytokines IL-6 and IL-12 [22].

Overall, there is limited research on the role of *DAB2* in T cells. *DAB2* is expressed exclusively in FOX3P^+^CD4^+^CD8^−^ T cells, with FOX3P promoting *DAB2* expression by binding to its promoter [23]. *DAB2* knock out (KO) in T cells has no effect on the overall number of T_reg_ cells in vivo, however, adoptive transfer of the *DAB2* KO T_regs_ had diminished efficacy against colitis in vivo [23]. This suggests *DAB2* is not crucial for maintenance of T_reg_ populations but is involved in their function.

## 3. Role of *DAB2* as a Tumour Suppressor

### 3.1. Expression of DAB2 in Cancer

One of the initial studies which identified *DAB2*, found its expression was lost in 90% of ovarian cancer cell lines [1]. Loss of *DAB2* expression is now considered an early step in the initiation of ovarian tumorigenesis [24]. *DAB2* is mainly expressed in the ovary, brain, kidney and intestine and its downregulation has been observed in cancers including those of the ovary, breast, lung, bladder, prostate, cervix and stomach (summarised in Table 1). In ovarian cancer, the percentage of tumours with positive staining for *DAB2* ranged from 0–26% [6,24,25,26]. Interestingly, 100% of mucinous tumours maintained *DAB2* staining [25], suggesting *DAB2* expression and function may vary between different ovarian cancer subtypes. This is also relevant in lung cancer where *DAB2* expression has been suggested as a marker for epithelioid mesothelioma which has 80–98% *DAB2* positivity compared to 3–23% for pulmonary adenocarcinoma cases [27,28]. Loss of the *DAB2* p96 isoform was observed in breast cancer, but low expression of the p67 isoform was observed in both normal and cancerous breast tissue [5,29]. Approximately 25% of lung cancer patients had high *DAB2* tumour expression compared to 56% in normal lung tissues [7,30]. *DAB2* gene expression was also decreased in cervical, gastric and prostate cancer compared to normal controls [31,32,33]. Research into pancreatic cancer found no *DAB2* expression in normal pancreatic tissues (n = 5), however in 7/8 pancreatic cancer tissues had low or high *DAB2* expression [34].

*DAB2* expression has also been associated with patient outcome in various cancers. In oesophageal cancer, low *DAB2* expression was associated with reduced overall survival (OS), increased risk of recurrence, larger tumour size, advanced stage of disease and metastasis [35]. In lung cancer, low *DAB2* protein expression was associated with reduced OS, reduced progression free survival (PFS), higher tumour stage and metastasis [30]. Low *DAB2* gene expression was also associated with reduced OS and PFS in patients with non-small cell lung carcinoma (NSCLC) [36]. In urothelial carcinoma of the bladder (UCB), low *DAB2* expression have been associated with high clinical stage and lymph node metastasis [37]. Another study was contradictory, finding high *DAB2* was associated with high clinical grade and reduced OS and PSF [38].

**Table 1 ijms-24-00696-t001:** Expression of *DAB2* in different cancers.

Cancer Type	*DAB2* Expression	Ref
Breast	*DAB2* p96 downregulated in cancer	[5]
	*DAB2* p67 low expression in normal and cancer tissue	[29]
Ovarian	*DAB2* downregulated in serous ovarian cancer	[25]
	*DAB2* maintained in mucinous ovarian cancer	[25]
	*DAB2* downregulated in ovarian cancer	[6,26]
	*DAB2* downregulated in serous, adenocarcinoma and mucinous cancer	[24]
Choriocarcinoma	*DAB2* increasingly downregulated from normal placental tissue, to partial mole, complete mole and choriocarcinoma	[39]
Urothelial Carcinoma of the Bladder (UCB)	*DAB2* downregulated in UCB Decreased *DAB2* expression associated with poor patient prognosis	[37]
Urothelial Carcinoma of the Bladder (UCB)	High *DAB2* expression associated with poor patient prognosis	[38]
Lung	Low *DAB2* gene and protein expression associated with significantly reduced PFS and OS	[40,41]
	Low *DAB2* associated with poor differentiation, higher tumour stage and lymph node metastasis	[40]
	*DAB2* gene and protein expression downregulated in cancer	[7,30,40,41]
	Methylation of *DAB2* promoter increased in cancer (93%) versus normal (35%)	[40]
Oesophageal squamous cell carcinoma (ESCC)	*DAB2* downregulated in cancer	[35,42]
	Low *DAB2* expression associated with poor patient prognosis	[35]
Cervical	*DAB2* downregulated in cancer	[31]
Gastric	*DAB2* downregulated in cancer	[32]
	*DAB2* downregulated in metastatic vs. primary tumours	[43]
Pancreatic	*DAB2* upregulated in cancer	[34]
Prostate	*DAB2* downregulated in cancer	[33]
Nasopharyngeal Carcinoma (NPC)	*DAB2* downregulated in cancer	[44]

### 3.2. Mechanisms for DAB2 Deregulation in Cancer

#### 3.2.1. Methylation of *DAB2* Promoter

The *DAB2* gene contains a CpG island at the 5′ end, suggesting promoter methylation may be one of the mechanisms regulating its expression [45]. A relationship between loss of *DAB2* expression and promoter methylation status has been observed in cancers of the lung [30,40], nasopharynx [44], head and neck [45,46], vulva [45] and liver [47]. *DAB2* promoter methylation was associated with poor cisplatin response and poor OS and PFS in squamous cell carcinomas (SCC) of the head and neck as well as the vulva [45]. Methylation of *DAB2* promoter was significantly increased in hepatocellular carcinoma patients with OS survival less than 3 years [47]. The relationship between *DAB2* expression and promoter methylation was not consistent for all cancer types. In NSCLC, *DAB2* expression was lost in 95% of primary tumours compared to matched normal tissues, with 85% of tumours having a higher methylation status of the *DAB2* promoter [48]. Another study in breast cancer found only 11% of patients had hypermethylation of the *DAB2* promoter despite 74% of patients exhibiting loss of *DAB2* expression [5]. In ESCC only 20% (n = 10) of patients with no *DAB2* expression had hypermethylation of the *DAB2* at the exon 1 promoter [42]. Another ESCC study found only 29% of patients with low *DAB2* expression had promoter hypermethylation [35]. Together these findings indicate that methylation may be responsible for loss of *DAB2* expression in some cancers. Re-expression of *DAB2* through targeting DNA methylation presents a possible treatment mechanism in tumours where methylation downregulation occurs.

#### 3.2.2. Translational Regulation of *DAB2* Expression

Heterogenous nuclear ribonucleoprotein E1 (HnRNPE1) has been implicated in the regulation of *DAB2* translation. hnRNPE1 binds to TGFβ activated translational (BAT) elements in the 3′UTR region of *DAB2* mRNA, preventing its translation [49]. Activation of TGFβ signalling promotes phosphorylation of hnRNPE1 at Ser43, preventing binding to the BAT elements in *DAB2* which promotes epithelial mesenchymal transition (EMT) in NMuMG and EpRas mammary epithelium [49]. Another mechanism for regulating *DAB2* expression is through the GATA6 transcription factor, which has been shown to directly enhance *DAB2* expression in transitional cell carcinomas (TCC) [50]. Loss of GATA6 was suggested as a precursor to pre-oncogenic transformation of serous, clear cell and endometrioid ovarian tumours but not mucinous ovarian tumours [51]. Furthermore, Mok et al. found that 100% of mucinous tumours were positive for *DAB2* expression [25]. This suggests a potential relationship between loss of GATA6 and *DAB2* expression in initiation of ovarian cancer. However, interestingly, this was not consistent in all ovarian cancer studies. Liu et al. found GATA6 to be significantly increased in high grade serous ovarian cancer (HGSOC) compared to non-serous subtypes (endometrioid, mucinous, clear cell, mixed, undifferentiated, malignant mixed mullerian tumour (MMMT)) [36]. Furthermore, in their study, HGSOC patients with GATA6 positive tumours had significantly reduced OS [36].

#### 3.2.3. *DAB2* Phosphorylation

Phosphorylation is another proposed mechanism for regulating *DAB2* activity in the cells. *DAB2* protein has 4 protein kinase C (PKC) phosphorylation sites at Ser^24^, Ser^32^, Ser^241^ and Ser^249^. PKC mediated phosphorylation of Ser^24^ inhibits the activity of AP-1 transcription factor [52]. The AP-1 transcription factor family includes c-Jun [53], c-FOS [44] and ATF which have previously been shown to be inhibited by *DAB2* [54]. During mitosis *DAB2* undergoes phosphorylation by cyclin-dependent kinase (cdc2) which causes it to dissociate from both the cell membrane and clathrin [55,56]. This inhibits *DAB2* mediated endocytosis resulting in cell arrest [56].

#### 3.2.4. MicroRNA Regulation of *DAB2*

Du et al. used TargetScan and miRmate programs to identify potential microRNAs that bind the 3′UTR of *DAB2* and downregulate its translation. They identified 9 microRNAs, including miR-93, miR-145, miR-26a, miR-26b, miR-124, miR-187, miR-203 and miR-153 [41]. Expression analysis of microRNAs and *DAB2* in lung cancer samples (n = 245) found a significant correlation between *DAB2* and miR-93 expression [41]. Additionally, *DAB2* overexpression significantly reduced cell proliferation via decreased Akt phosphorylation which was inhibited by overexpression of miR-93 [41]. This was consistent in acute myeloid leukaemia cells, where miR-93 downregulation enhanced *DAB2* expression and subsequently enhanced cell apoptosis and reduced cell proliferation and in vivo tumorigenesis [57].

EMT has been indicated as a key process in the metastasis of tumours [58]. It is a conversion from an epithelial to a mesenchymal phenotype which is associated with loss of adherin molecules and a complex signature of transcription factors. TGFβ can promote EMT in cancer [59]. miR-106b has been shown to enhance TGFβ1 mediated migration in cervical cancer cell lines HeLa and SiHa [31,60]. miR-106b was also shown to enhance proliferation and migration of hepatocellular carcinoma (HCC) cells, which was inhibited by *DAB2* expression [61]. miR-106b binds the 3′UTR of the *DAB2* mRNA and there is a negative relationship between *DAB2* and miR-106b expression [61]. miR-106b is part of novel microRNA cluster also consisting of miR-93 and miR-25 [62]. This microRNA cluster was shown to promote the switch from TGFβ growth suppression to TGFβ mediated EMT in MCF-7 breast cancer cells [63]. miRNA mediated *DAB2* loss has also been reported in Epstein–Barr virus-associated gastric cancer by miR-BART1-3p [64]. miR-134-5p expression in stage I lung cancer was associated with early relapse [65]. miR-134-5p could silence *DAB2* expression and was associated with reduced E-cadherin expression and enhanced migration, invasion, in vivo metastasis and resistance to cisplatin in lung cancer cell lines [65]. Oestrogen-induced cell proliferation was associated with increased miR-191 expression and the silencing of *DAB2* expression in oestrogen receptor (ER) positive breast cancer cell lines [66]. Inhibiting miR-191 expression in ER+ breast cancer, enhanced *DAB2* expression and reduced tumorigenesis in vivo [66]. 17β-estradiol enhanced expression of miR-378 in mouse ovarian surface and fallopian epithelium which was associated with a rapid and significant reduction in *DAB2* expression and cell dysplasia [67]. *DAB2* was reported to be a target of miR-145 [60,68]. Epigallocatechin gallate increased miR-145 in rat cardiomyocytes which in turn repressed *DAB2* expression [69]. miR-149 also downregulates *DAB2*, activating Wnt signalling in mouse bone marrow derived mesenchymal stem cells [70].

## 4. *DAB2*, a Negative Regulator of Pro-Tumorigenic Signalling Pathways

There is a complex network of extracellular and intracellular signalling pathways which are key to the progression of cancer. EMT is considered a key process involved in the progression and metastasis of cancer [71]. In the absence of *DAB2*, there is activation of key pro-tumorigenic and pro-EMT signalling pathways including the MAPK, Wnt/β-catenin and TGFβ pathways (Figure 2). The functional outcomes of *DAB2* signalling in cancer is summarised in Table 2.

### 4.1. Activation of ERK/MAPK Signalling

Mitogen activated protein kinase (MAPK) signalling pathway involves binding of a range of stimuli including growth factors, cytokines and mitogens to a G-protein coupled receptor [76]. This activates a signalling cascade via mitogen-activated protein kinase kinase kinase (MAPKKK) which in turn activates MAPKK and then MAPK, promoting expression of target genes which leads to cell proliferation, differentiation, and migration [76]. In cancer, activation of MAPK signalling pathways is associated with chemotherapy resistance [77] and metastasis [78]. The extracellular signal-regulated kinase (ERK) family of MAPK are deregulated in approximately one third of human cancers [76,79]. Activation of ERK is initiated upon binding of a growth factor or mitogen to a receptor tyrosine kinase. Growth factor receptor-binding protein 2 (Grb2) is recruited and activated triggering a sigalling cascade of serial phosphorylation of different kinases from son of sevenless (SOS), to Ras, Raf, MEK/2 and finally ERK1/2 [76,79] (refer to Figure 2).

*DAB2* inhibits the ERK/MAPK signalling pathway by disrupting the interaction between SOS and Grb2 [2]. *DAB2* PRD interacts with the SH3 domains of Grb2, c-Src and Fg r [80]. *DAB2* PRD can bind both the C-and N-terminal SH3 domains of Grb2 preventing the interaction of SOS with either SH3 domain [2,5,81]. Downregulation of *DAB2* enhances free Grb2 for binding to SOS, activating the MAPK signalling pathway. Active ERK1/2 has been reported to promote EMT in two mammary epithelial cell lines (MCF10A1 and HME5-cdk4) [29]. EGF activated Erk2 via phosphorylation and activation of upstream c-Src at Tyr-416. The *DAB2* PRD inhibited this activation by binding the SH3 domains within c-Src and subsequently inhibiting Erk2 [80]. Retinoic acid (RA) was shown to inhibit Erk1 activation in F9 embryonic stem cells which further inhibited activation of Elk-1 and c-Fos transcription [82]. RA treatment also enhanced *DAB2* expression in F9 cells which was hypothesised as the mechanism for Erk1 inhibition [82]. Interestingly, *DAB2* is a target of MAPK mediated phosphorylation [80]. *DAB2* inhibition of ERK1/2 and c-Fos was confirmed in ovarian (OVCAR3, PA-1) and breast cancer cell lines (MCF-10, SK-Br-2 and MCF-7) [6]. *DAB2* overexpression promotes cell death in ovarian cancer cells (OVCAR3) in normal tissue culture conditions, however when grown on a basement membrane, the inhibitory effect of *DAB2* overexpression was reversed [6].

### 4.2. Activation of Wnt/β-Catenin Signalling

The Wnt signalling pathway is involved in cell proliferation, embryonic development, cell motility, differentiation, stem cell signalling and invasion [83,84]. Deregulation of Wnt signalling has been reported primarily in colorectal cancer but also in pancreatic and liver cancer [83,84]. Activation of the Wnt signalling pathway occurs by canonical and non-canonical pathways [83,84]. Canonical Wnt activation requires translocation of β-catenin to the nucleus where it activates its primary targets, cyclin D1 and c-myc [83,84]. Inactive Wnt signalling is maintained by a β-catenin destruction complex composed of Axin, glycogen synthase kinase 3 (GSK3), adenomatous polyposis coli (APC) and casein kinase I (CKI) which bind β-catenin and phosphorylate it, thereby targeting it for ubiquitinase mediated degradation [83,84]. Binding of Wnt ligand to the receptors frizzled and LRP5/6 promotes recruitment of Dishevled (Dvl) which disrupts the destruction complex, saving β-catenin from digestion [83,84].

*DAB2* is a key regulator of Wnt signalling in differentiation of human embryonic stem cells into cardiomyocytes [85]. Expression of *DAB2* and β-catenin are negatively correlated [30]. *DAB2* overexpression and knockdown were associated with reduced and enhanced β-catenin expression, respectively, in lung and gastric cancer cells [7,30,43] and NIH-3T3 mouse fibroblasts [86]. A positive correlation between *DAB2* and Axin expression was also observed in LK2 NSCLC cells [30]. Further analysis of the interactions between *DAB2* and Wnt signalling components has demonstrated a direct interaction between *DAB2* PTB and the Dvl-3 DEP domain and the Axin N-terminal region (aa 194-956) [86]. Upon Wnt3A signalling activation, Dvl-3 and Axin interact directly via their PDZ and N-terminal domains, respectively disrupting the β-catenin destruction complex. This allows nuclear translocation of β-catenin and transcription of targets c-Myc and cyclin D1 [86,87]. *DAB2* binds both Dvl-3 and Axin, disrupting their interaction, maintaining the destruction complex and allowing glycogen synthase kinase 3β (GSK3β) phosphorylation and subsequent ubiquitin mediated digestion of β-catenin [86]. Axin is crucial for maintaining inactive Wnt signalling by stabilising the destruction complex and maintaining in-active Wnt signalling [88]. Phosphorylation of Axin is important for its own stability. Upon canonical Wnt activation, LRP5/6 is phosphorylated upon dimerisation with frizzled. This promotes recruitment of Axin which in turn is de-phosphorylated by protein phosphatase 1 (PP1) [88]. The destabilisation of Axin is further regulated by *DAB2* [89]. *DAB2* prevents the interaction between Axin with both PP1 and LRP5 [89]. PP1 and *DAB2* bind to the C-terminal end of Axin indicating a competitive binding relationship [89]. The inhibition mechanism of Axin-LRP5 interactions by *DAB2* is not known, but as the *DAB2* PTB binds the FXNPXY sequence in other LDL family members, a direct interaction is hypothesised.

*DAB2* has been shown to regulate Wnt signalling through direct interactions between *DAB2* PTB domain and the intracellular domain of LRP6 [90]. In the absence of *DAB2*, Wnt3A signalling activates LRP6 via phosphorylation by GSK3β, LRP6 then undergoes calveolin mediated endocytosis and interacts with Axin, activating β-catenin signalling [91]. In the presence of *DAB2*, Wnt3A signalling promotes casein kinase 2 (CK2) phosphorylation of LRP6 at S1579A. This promotes binding of *DAB2* to LRP6 and association with clathrin, thereby inhibiting the interaction of LRP6 with Axin and in turn inhibiting Wnt signalling resulting in reduced in vivo tumorigenesis as described in F9 teratocarcinoma cells [90]. In SGC gastric cancer cells, knockdown of *DAB2* expression was associated with enhanced cell migration and enhanced expression of Wnt signalling components, including β-catenin, GSK3β and cyclinD1 [43].

Non-canonical Wnt signalling occurs independently of β-catenin and one of its primary targets is the planar cell polarity (PCP) signalling cascade which activates downstream Jun-N-terminal kinase (JNK) [92]. *DAB2* has also been shown to regulate the non-canonical PCP-PE pathway [53,86]. *DAB2* enhances Dvl1-3 activation of JNK via Wnt-5A signalling [86]. *DAB2* has been shown in inhibit cholesterol-dependent activation of JNK and c-Jun by TGFβ1 through sequestering of TGFβRI [53].

### 4.3. Regulation of TGFβ Signalling Pathways

TGFβ is a key regulatory cytokine which is recognised by most human cells. TGFβ is responsible for maintaining normal homeostasis and in turn has tumour suppressive function [93]. Despite this, tumour cells are capable of evading TGFβ signalling and also utilising TGFβ signalling for their own benefit [94]. The TGFβ superfamily comprises over 30 members and has both canonical and non-canonical pathways, indicating a vast and complex network with diverse biological implications [93,94]. The principal mechanism of canonical TGFβ signalling involves binding of TGFβ ligand to a type II receptor, a serine threonine kinase which then recruits and phosphorylates a type I receptor [93]. Transcription factor SMAD is then phosphorylated and interacts with multiple other transcription factors to elicit and range of signals [93].

TGFβ signalling is complex and can act as both a tumour suppressor and a tumour promoter. Loss of *DAB2* expression in tumour compared to normal tissues is well documented (Table 1). A particular study in head and neck and vulval human squamous cell carcinoma (HSCC) suggested loss of *DAB2* expression acts as a switch for TGFβ signalling to change from a tumour suppressor to a tumour promoter role [45]. They found that with *DAB2* expression, TGFβ activation of Smad2 and subsequent cell proliferation and motility were reduced. However, loss of *DAB2* expression enhanced TGFβ Smad2 activation, reverting the effects on cell proliferation and enhanced cell motility [45]. A study in EpH4 mammary epithelium overexpressing Ras cells (EpRas), suggested cross talk between MAPK and TGFβ signalling in promoting EMT and tumour metastasis [95]. Another study in mammary epithelial cell lines (MCF10A1 and HME5-cdk4) demonstrated that *DAB2* knockdown promoted activation of ERK which enhanced expression of TGFβ2 and promoted an EMT phenotype observed by reduced E-cadherin and enhanced N-cadherin and vimentin [29]. Hocevar et al. suggests that the tumour suppressive effects of *DAB2* may occur through dual inhibition of more than one key signalling pathway [96]. In pancreatic cancer cell lines (COLO357 and PANC), *DAB2* knock down enhanced TGFβ-mediated EMT through reduced E-cadherin and enhanced Snail, Slug and N-cadherin expression [96]. A functional mechanism for *DAB2* in maintaining an epithelial phenotype is through co-localisation with E-cadherin at the plasma membrane, maintaining apical junctions [97]. Loss of *DAB2* is associated with cytosolic localisation of E-cadherin and β-catenin [97].

TGFβ signalling can be activated through endocytosis, both clathrin and calveoli mediated, although Smad activation also occurs independently of endocytosis [98]. *DAB2* was shown to regulate TGFβ clathrin mediated endocytosis of TGFβRI in ES-2 ovarian cancer cells [53]. *DAB2* was not involved in the endocytosis of TGFβRII in NIH/3T3 mouse fibroblast, but it did mediate the intracellular trafficking of TGFβRII, particularly the transfer of TGFβRII from EEA1- positive early endosomes to Rab11-positive recycling endosomes [99]. *DAB2* may also modulate TGFβ signalling through directly interacting with Smad effector proteins when TGFβ is impaired through loss of TGFβRI/II [100]. *DAB2* PTB interacts with the MH2 domain of Smad2 and Smad3 but not Smad1 or 4 [100]. *DAB2* expression is sufficient to recover TGFβ signalling and subsequently enhances phosphorylation of Smad2 and nuclear translocation of both Smad2 and Smad3 [100].

## 5. The Role of *DAB2* as a Tumour Promoter

### 5.1. DAB2 and TGFβ Pro-Tumorigenic Signalling

There is strong evidence describing the tumour suppressive function of *DAB2* particularly through inhibition of key signalling pathways involved in cell survival and cell fate determination. As shown in Table 1, the majority of cancer tissue have either a complete loss or significant reduction in *DAB2* expression which has been associated with malignant transformation of cells. Despite most of the evidence supporting *DAB2* is a tumour suppressor, some studies have found contradictory findings suggesting a tumour promoting role for *DAB2* (Table 2). There is a strong relationship between *DAB2* and TGFβ signalling [75]. It is well documented that TGFβ functions as both a tumour suppressor and tumour promoter [94]. In line with this, some studies have demonstrated that TGFβ together with *DAB2* can have pro-tumorigenic effects [35,73,75].

TGFβ treatment in normal murine mammary gland epithelium (NMuMG) cells enhanced EMT and cell survival via *DAB2* [75]. TGFβ via *DAB2* enhanced activation of focal adhesion kinase (FAK) which in turn activated β1 integrin, preventing apoptosis [75]. Inhibition of *DAB2* prevents TGFβ mediated EMT, through loss of N-cadherin and enhanced cell apoptosis [75]. Another study in ESCC found a negative correlation between *DAB2* expression and cell migration associated with ERK activation [35]. Interestingly, when treated with TGFβ1 an EMT phenotype (enhanced CDH2, reduced CDH1) was observed in KYSE50 cells with high but not low *DAB2* expression [35]. Activation of TGFβ signalling promoted phosphorylation of hnRNPE1 at Ser43, preventing binding to the BAT elements in *DAB2* and ILEI which promoted *DAB2* translation and EMT in NMuMG and EpRas mammary epithelium [49]. TGFβ enhanced expression of *DAB2* and promoted localisation to the membrane and interacted with β1 integrin [75]. TGFβ via *DAB2* promoted the activation of FAK kinase, activating β1 integrin, promoting cell survival [75]. *DAB2* regulates internalisation of free and inactive integrin β1 in a clathrin independent mechanism and traffics integrin β1 to perinuclear recycling endosomes. Integrin β1 is then returned to the cell surface, where it interacts with vinculin (focal complex protein) enhancing migration of HeLa cancer cells [101].

### 5.2. DAB2 Promotes EMT and Metastasis

Another study demonstrated that *DAB2* correlates with metastatic potential in the two prostate cancer cell lines, PC3 and LNCaP [73]. *DAB2* overexpression was shown to promote migration and invasion in LNCaP cells and was associated with expression of migration associated genes, whereas knock down of *DAB2* in the more metastatic PC3 cells had opposing effects on cell invasion and migration [73]. *DAB2* expression was also associated with tumour progression in urothelial carcinoma [38]. High *DAB2* expression associated with reduced PFS and OS, particularly in metastatic urothelial carcinoma which had invaded the muscle [38]. siRNA mediated knockdown of *DAB2* expression in UM-UC-3 bladder cancer cells was associated with reduced tumour formation in vivo and subsequent enhanced expression of EMT marker *KRT14* and reduced expression of the mesenchymal to epithelial transition (MET) marker occludin (*OCLN*) [38]. Secreted factors from *DAB2* expressing stromal cells also promoted expression of EMT markers in Um-UC-3 cells [38]. *DAB2* was also suggested to promote EMT in ovarian cancer [72]. Chao et al. found miR-187 to be associated with poor survival in ovarian cancer patients [72]. They found similar to other studies that miR-187-reduced *DAB2* expression via the 3′UTR and was associated with reduced cell proliferation [72]. miR-187 overexpression in SKOV3 ovarian cancer cells promoted MET expression patterns including enhanced E-cadherin and reduced vimentin and phospho-FAK [72]. This was associated with reduced cell migration which could be reversed by overexpression of *DAB2* [72]. In choriocarcinoma cell lines with similarly low endogenous *DAB2* protein levels, *DAB2* overexpression results in a 15%, 64% and 86% reduction in cell growth in BeWo, Jar and JEG choriocarcinoma cell lines, respectively, indicating that *DAB2* expression can only inhibit cell proliferation in certain cell types [39]. Loss of *DAB2* has been shown to inhibit cell migration of fibrosarcoma HT1080 cells as it is involved with AP-2 in the disassembly of the focal adhesion complex [102].

A small study in pancreatic cancer found *DAB2* expression was enhanced in tumour versus normal pancreatic tissues. However, in metastatic tissues *DAB2* expression was found to be reduced [34]. In grade 3, stage 1 (T1G3) bladder cancer there was a 2.85 fold increase in *DAB2* expression in patients that progressed versus patients that did not [103]. *DAB2* positivity for epithelioid mesothelioma was 80–98% compared with 3–23% inpulmonary adenocarcinoma [27,28]. *DAB2* has therefore the potential to be used a novel marker for differentiating these two subtypes of lung malignancies.

### 5.3. Tumour Associated Macrophages (TAMs)

Recent research suggests *DAB2* has tumour promoting effects in the tumour microenvironment (TME). *DAB2* was highly expressed in TAMs and its knockdown significantly reduced lung metastasis in mouse fibrosarcoma and breast cancer models [104]. Immune cells elicit a range of pro-tumorigenic and tumour suppressive effects [105]. In particular, TAMs are well studied. M1 macrophages are thought to have a predominantly tumour suppressive function by actively promoting inflammation and directly targeting cancer cells [105]. M2 macrophages however are considered to have tumour promoting function as they enhance angiogenesis within the TME, and release a range of pro-metastatic secretory factors [105]. M2 macrophages have also been shown to enhance cell invasion and motility in various cancers such as those of the breast, stomach and lung [19,106]. A study by Adamson et al. demonstrated that *DAB2* is involved in the polarisation of macrophages to a M2 phenotype, whereas *DAB2*-silencing promoted a pro-inflammatory M1 phenotype in both mouse and human bone marrow-derived macrophages (BMDM) [19]. High *DAB2* macrophages in peritumoral and intratumoral areas was shown to associate poorly with disease free survival, lymph node metastases and tumour cell proliferation in breast cancer patients [104]. Colony stimulating factor-1 (CSF-1) also promotes an M2 phenotype [107]. CSF-1 enhanced *DAB2* expression in TAMs isolated from MN-MCA1 fibrosarcoma mice models only in adherent conditions [104]. This was found to occur via the mechanotransductive YAP-TAZ transcription complex [104]. *DAB2* KD in myeloid cells significantly reduced the invasiveness of E0771 breast cancer cells in vitro as well as number of lung metastases in vivo [104]. This was found to be a result of reduced internalisation and recycling of integrins and ECM components including collagen I, collagen IV, fibronectin and laminin [104]. Interestingly, PD-L inhibitors, which allow T-cell mediated death of tumour cells, further reduced the number of lung metastases in *DAB2* KO mice but not WT mice [104]. This highlights a potential need to explore *DAB2* inhibitors in combination with PD-L inhibitors in treating advanced staged cancers.

### 5.4. Regulation of Angiogenesis

Throughout tumour initiation, progression and metastasis, tumours cells release a range of factors which modulate their microenvironment [108]. The development of the TME includes the formation of new blood vessels which supply necessary oxygen and nutrients to the growing tumour [108,109]. TGFβ signalling is an important signalling pathway in angiogenesis, not only in the TME but also in embryogenesis [93]. *DAB2* promotes TGFβ1 mediated expression of the angiogenic factors VEGF and FGF-2 and also activation of MAPK signalling, particularly phosphorylation of ERK [110]. Angiogenesis requires VEGFR signalling which is activated through internalisation of receptors VEGFR2 and VEGFR3 [111,112]. Ephrin-B2 is a key protein for VEGFR2 internalisation but it also requires PAR-3 and *DAB2* which interacts directly with VEGFR2/3 via the PTB domain [111,112,113]. In mature vessels VEGFR2/3 internalisation is inhibited by atypical protein kinase C (aPKC) which phosphorylates the *DAB2* PTB domain, preventing its interaction with VEGFR2/3 [114]. Syndecan-1 activates aPKC and subsequent phosphorylation of *DAB2* inhibiting its activation of VEGF-VEGFRII signalling [114]. *DAB2* promotes internalisation of VEGFRI and VEGFRII in liver sinusoidal endothelial cells which is required for their dedifferentiation, proliferation and migration during angiogenesis [115].

## 6. Targeting *DAB2*

Our review highlights how *DAB2* in the majority of cases is associated with tumour suppressive phenotypes and hence is commonly downregulated. Re-expression of *DAB2* in these cases should therefore be tumour suppressive and a potential treatment strategy. As discussed, there is growing evidence of the role of miRNAs targeting and downregulating *DAB2*. Targeting miRNAs in cancers and subsequent re-expression of *DAB2* may be a suitable co-treatment with other current treatment strategies. Green tea has previously been shown to have anti-tumorigenic effects [50]. Yang et al. isolated crude polysaccharide from green tea which enhanced apoptosis in PC-3 prostate cancer cells [51]. They demonstrated the polysaccharide downregulated miR-93 which in turn enhanced *DAB2* expression, activating both ERK and PI3K [25,51]. Re-expression of *DAB2* through targeting DNA methylation presents another possible treatment mechanism in tumours where methylation downregulation occurs. In NSCLC cell line, LK2, x-ray irradiation promoted de-methylation of *DAB2* CpG sites and enhanced *DAB2* and Axin expression which inhibited Wnt signalling, cell proliferation and in vivo tumour formation [22]. *DAB2* is important in regulating membrane integrity, immune regulation and key signalling pathways including MAPK, PI3K and Wnt and therefore will likely be difficult to directly target. A greater understanding of functional role of *DAB2* in the TME may highlight potential treatment strategies to target *DAB2* in cancer.

## 7. Conclusions

*DAB2*, initially identified as a tumour suppressor, is also an important adaptor molecule for clathrin-mediated endocytosis. Although the majority of literature has demonstrated either downregulation or loss of *DAB2* expression in tumour tissues compared to normal tissues there are also several studies which have demonstrated an increase in *DAB2* expression in cancer and an association with tumour progression. Additionally, some studies have shown that loss of *DAB2* expression is not consistent amongst all subtypes of a particular cancer type. The majority of functional studies demonstrate that *DAB2* is a negative regulator of the key signalling pathways Wnt, MAPK and TGFβ which elicit pro-tumorigenic effects. Some research contradicts these findings and describe a contrary effect, in particular through regulation of the TGFβ pathway which is known to be both pro-tumorigenic and tumour suppressive. *DAB2* contains multiple binding domains and therefore has the ability to interact with multiple proteins simultaneously, explaining how different cell conditions may impact its function. More recent research has identified miRNA expression as a mechanism by which *DAB2* is downregulated. Treatment strategies against pro-tumorigenic miRNAs are rapidly evolving offering a potential mechanism for re-activating *DAB2* expression. A greater understanding of the functional role of *DAB2* in the TME may lead to the development of novel strategies to block tumour progression.

## Figures and Tables

**Figure 1 ijms-24-00696-f001:**
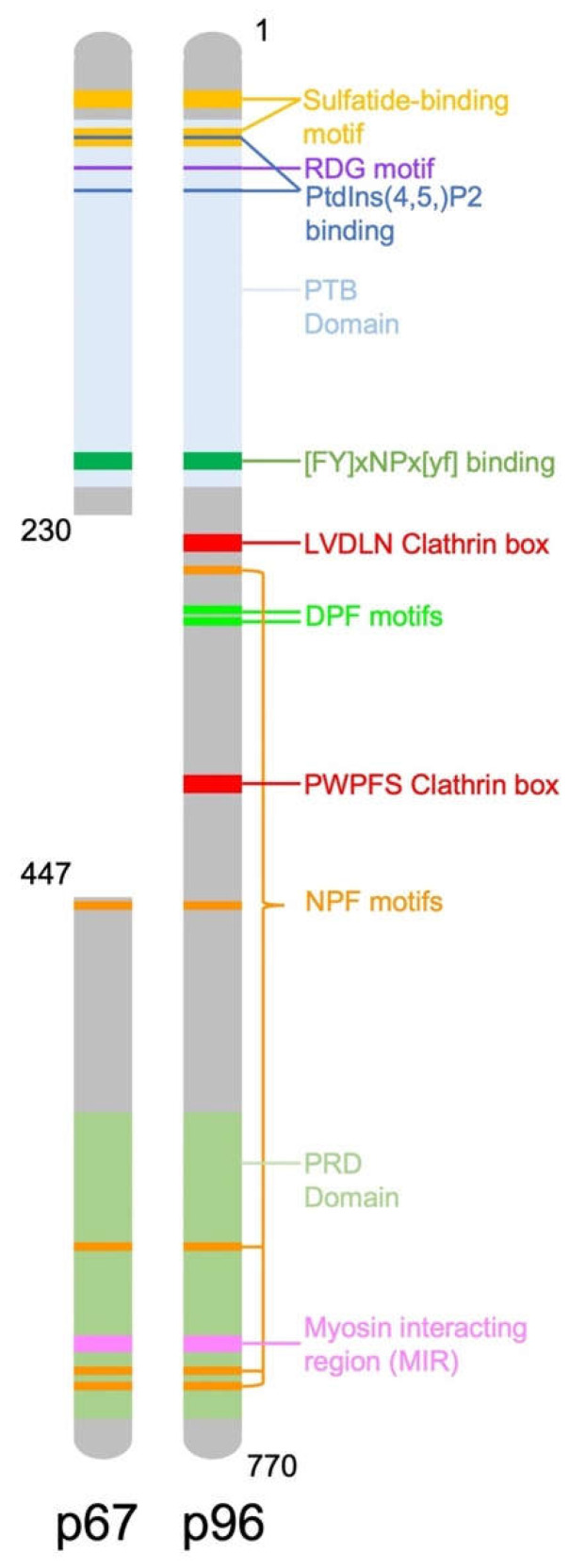
Diagram representing the structure of the *DAB2* protein and location of important functional domains.

**Figure 2 ijms-24-00696-f002:**
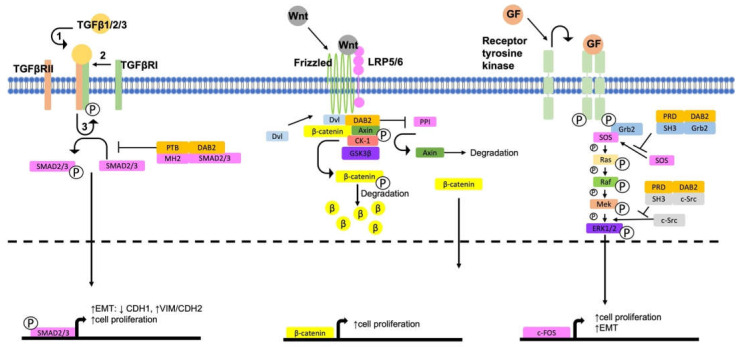
Regulatory role of *DAB2* in MAPK, Wnt/β-catenin and canonical TGFβ pathways. In the absence of *DAB2*, TGFβ1/2/3 activate canonical signalling pathways through dimerisation of receptors TGFβRI and TGFβRII. This activates SMAD2 and enhances cell proliferation and EMT through expression of target genes. *DAB2* PTB domain directly interacts with the MH2 domain of SMAD2/3, preventing its activation. *DAB2* inhibits the canonical Wnt signalling pathway by preventing the interaction between Dvl and Axin, preventing destruction of the β-catenin destruction complex. In the absence of *DAB2*, Dvl and Axin interact separating the complex. β-catenin is then free to translocate to the nucleus where it promotes cell proliferation through expression of target c-myc and cyclin D1. *DAB2* PRD interacts with the SH3 domains of SOS and c-Src in the MAPK pathway, inhibiting the signal transduction and in turn inhibiting cell proliferation and EMT.

**Table 2 ijms-24-00696-t002:** The functional roles of *DAB2* in cancer.

Cancer Type	Observation	Role	Ref
**Cell proliferation**			
Acute Myeloid Leukaemia	Knockdown of miR-93 enhances *DAB2* expression and inhibits cell proliferation in THP-1 cells in vitro and in vivo	TS *	[57]
Hepatocellular carcinoma (HCC)	miR-106b knockdown of *DAB2* enhances Hep3B cell proliferation in vitro	TP **	[61]
Lung adenocarcinoma	Knockdown of *DAB2* inhibits A549 and H1299 cell growth and overexpression of *DAB2* enhance A549 and H1299 cell growth	TP	[65]
Breast cancer	Oestrogen enhances miR-191 and silences *DAB2* expression and promotes cell proliferation in ER positive breast cancer	TS	[66]
Head and Neck and Vulval Squamous cell carcinoma (SCC)	TGFβ inhibits cell proliferation in cell lines (HN30, H376, H413, Procotor, UMSCV1A, UMSCV1B and UMSCV7) that have high levels of *DAB2*	TS	[45]
Urothelial Carcinoma of the Bladder (UCB)	Downregulation of *DAB2* decreases the proliferation of UM-UC3, J82 and T24 cells	TP	[38]
Ovarian cancer	miR-187 in SKOV-3 cells suppressed *DAB2* expression and enhanced cell proliferation	TS	[72]
**Migration**			
Hepatocellular carcinoma (HCC)	miR-106b knockdown of *DAB2* enhances Hep3B cell migration	TS	[61]
Head and Neck and Vulval SCC	TGFβ inhibits cell motility in cell lines (HN30, H413, UMSCV1A, UMSCV1B and UMSCV7) that express high levels of *DAB2*	TS	[45]
Lung adenocarcinoma	Silencing *DAB2* enhances cell migration in A549 and H1299 cells in vitro and overexpression of *DAB2* reduced cell migration in A549 and H1299 cells	TS	[65]
Prostate cancer	*DAB2* overexpression enhanced LNCaP cell migration and *DAB2* knockdown by shRNA inhibited PC3 cell migration	TP	[73]
Urothelial UCB	Downregulation of *DAB2* decrease the migration of UM-UC3 and T24 cells	TP	[38]
Ovarian cancer	miR-187 suppressed *DAB2* expression and inhibited cell migration in SKOV-3 cells	TP	[72]
Gastric cancer	Downregulation of *DAB2* promote SGC cell migration via Wnt/β-catenin and Hippo-YAP signalling pathways	TS	[43]
**Invasion**			
Prostate cancer	*DAB2* expression enhanced LNCaP cell invasion and *DAB2* knockdown inhibited PC3 cell invasion	TP	[73]
Urothelial UCB	Downregulation of *DAB2* decreased cell invasion of J82 and T24 cells	TP	[38]
**Apoptosis**			
Breast Cancer	*DAB2* promotes anoikis in SK-BR-3 and MDA-MB-453 cells	TS	[74]
Normal murine mammary gland (NMuMG)	Down regulation of *DAB2* enhance TGFβ induced apoptosis	TP	[75]
Breast cancer	*DAB2* sensitises SK-BR-3 and MDA-MB-453 cells to apoptosis by inhibiting the activity of integrin-linked kinas (ILK)	TS	[74]
**In vivo tumour growth and metastasis**			
Lung adenocarcinoma	*DAB2* is a target for miR-134-5p. Overexpression of miR-134-5p increased A549 cells tumour growth in mouse model	TS	[65]
Prostate cancer	*DAB2* knockdown inhibits PC3 cells tumour growth and metastasis in the mouse model	TP	[73]
Urothelial UCB	Reduced tumour growth and invasion in xenograft tumours of UM-UC-3 cells treated with *DAB2* targeting siRNA	TP	[38]
Ovarian	*DAB2* overexpression reduces SKOV3 tumour formation in nude mice	TS	[25]

* TS = tumour suppressor; ** TP = tumour promoter.

## Data Availability

Not applicable.

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
