# Peer review of "Disabled-2 (DAB2): A Key Regulator of Anti- and Pro-Tumorigenic Pathways"

_ijms, 2022, doi:10.3390/ijms24010696_

Round 1

Reviewer 1 Report

Disabled-2 (DAB2): a key regulator of anti- and pro-tumorigenic pathways

This review provides comprehensive literature data on the function of the tumour suppressor, DAB2. The review also covers the complex and multi-role of DAB2 in cancer.

The review discusses the two contradictory roles the DAB2 plays, on one hand, many studies showed that DAB2 downregulation or loss of expression in tumour tissue but on the other hand several studies demonstrated that an increase of DAB2 expression is associated with cancer and tumour progression.

The review could be considered as a good reference for the complicated role of DAB2 in cancer.

Author Response

We thank the reviewer for their kind comments on our manuscript 

Reviewer 2 Report

In this review, Price et al. summarized the regulation function of DAB2 in cancers. DAB2 was first identified as a suppressor in tumor cells, and recent studies demonstrated the pro-tumorigenic functions of DAB2 related to the immunosuppression of TAMs. Although the review lists the tumor-related studies about DAB2 in detail; however, most information has been published and there is no insight or perspective study in this manuscript. The authors should at least discuss the potential and strategies of targeting DAB2 for cancer therapy. In addition, the roles of DAB2 in immune regulation, such as APCs and Tregs, have been discussed recently (PMID: 33178208). The authors should include this information to comprehensively introduce DAB2 functions. In addition, I have some other comments listed below.

1. There are many literatures introducing mouse DAB2. Both human and mouse DAB2 gene structures and functions should be presented in figure 1 and section 2.

2. Please include the current knowledge on the roles of DAB2 in the innate and adaptive immune system

3. Table 2. The authors should discuss the DAB2 roles in tumor cells and TAMs separately. 

4. What’s the regulation mechanism of DAB2 in TAMs?

5. How to develop DAB2-related cancer therapy? Considering the dual anti- and pro-tumorigenic, will targeting DAB2 be a promising strategy to treat cancers?

Author Response

We thank you for your constructive comments. We have addressed your comments and suggestions. Please refer to the word document attached

Round 2

Reviewer 2 Report

No more comment.